# Transcriptional Modulation Reveals Physiological Responses to Temperature Adaptation in *Acrossocheilus fasciatus*

**DOI:** 10.3390/ijms241411622

**Published:** 2023-07-19

**Authors:** Zhenzhu Wei, Yi Fang, Wei Shi, Zhangjie Chu, Bo Zhao

**Affiliations:** 1College of Fisheries, Zhejiang Ocean University, Zhoushan 316022, China; weizhenzhu@zjou.edu.cn (Z.W.); shiwei@zjou.edu.cn (W.S.); chuzhangjie@zjou.edu.cn (Z.C.); 2Zhejiang Marine Fisheries Research Institute, Zhoushan 316021, China; fyjhsr@gmail.com

**Keywords:** *Acrossocheilus fasciatus*, temperature stress, RNA-seq, differentially expressed genes

## Abstract

In order to explore the molecular regulatory mechanism of temperature acclimation under long-term temperature stress in *Acrossocheilus fasciatus*, this study used high-throughput sequencing technology to analyze 60 days of breeding under five temperature conditions (12 °C, 16 °C, 20 °C, 24 °C, 28 °C). Compared with 20 °C, 9202, 4959 differentially expressed genes (DEGs) were discovered in low-temperature groups (12 °C, 16 °C), whereas 133 and 878 DEGs were discovered in high-temperature groups (24 °C, 28 °C), respectively. The KEGG functional enrichment analysis revealed that DEGs were primarily enriched in tight junction, PI3 K-Akt signaling pathway and protein digestion and absorption in low-temperature groups, and mainly enriched in proximal tubule bicarbonate reclamation, protein digestion and absorption, and HIF-1 signaling pathway in high-temperature groups. The viability of transcriptome sequencing-based screening of DEGs for temperature adaptation in *A. fasciatus* was shown by the selection of eight DEGs for further validation by quantitative real-time PCR (qRT-PCR), the findings of which were consistent with the RNA-seq data. According to the findings, protein digestion and absorption were primarily regulated by temperature variations, physiological stress was a significant regulator in regulation under high-temperature stress, and the immune system was a significant regulator in regulation under low-temperature stress. The transcriptional patterns of *A. fasciatus* under temperature stress are revealed in this study. This knowledge is crucial for understanding how *A. fasciatus* adapts to temperature and can help us better comprehend the environmental difficulties that *A. fasciatus* adaptation faces.

## 1. Introduction

*Acrossocheilus fasciatus* (Steindachner, 1892), which belongs to Cypriniformes, Cyprinidae, Barbinae, *Acrossocheilus*, is a lukewarm water fish. It is a commercial fish that is primarily found in mountain streams and rivers south of the Yangtze River in China [1]. In recent years, the natural resources of *A. fasciatus* have declined significantly due to weather, water pollution, excessive and illegal fishing, and other factors. *A. fasciatus* has been bred and propagated in captivity in an effort to preserve and replenish natural resources [2]. It has been demonstrated that the *A. fasciatus* can survive in a water temperature range of approximately 12–30 °C [3], which may be an excellent model for studying the response mechanism of stream fish to temperature.

The extreme temperature caused by global climate change constitutes a huge challenge to the *A. fasciatus* breeding industry. Temperature is one of the most significant environmental elements impacting fish life activities since it affects growth, development, reproduction, behavior, and metabolism [4,5]. A cohabitation model suggests that variations in water temperature may affect pathogen transfer and disease development in fish farming, which likewise faces major illness issues brought on by pathogenic bacteria and stress. Because variations in water temperature may impact the spread of disease and pathogens in a cohabitation model [6,7], resulting in reduced temperature adaptation. Environmental stress, especially temperature, can disrupt the organism’s internal balance and adversely affect its biological functions. Fish immune systems can be impacted by low temperatures, which can impair their ability to perform certain immunological functions and reduce fish survival [8]. Fish may experience altered protein synthesis and reduced glucose metabolism at high temperatures [9]. Fish’s metabolic rate, energy balance, and, consequently, their energy needs are impacted by temperature changes [10]. Fish can adjust to changes in ambient temperature by engaging in specific physiological processes. However, different fish have varying degrees of temperature adaptability [11]. Fish are capable of adapting to environmental temperature by modifying their physiological activity within a specific temperature range, which has become a popular research topic. For instance, when exposed to temperatures that are higher than those that are suitable for fish, fish will stabilize their metabolic rate to ensure a sufficient supply of oxygen to their tissues. Exposure to higher temperatures can also cause compensatory changes in the body’s energy reserve concentration [12], glycolysis, and enzyme activity in the mitochondria [13]. The peripheral nervous system and temperature receptors in the hypothalamus, as well as the central nervous system, all perceive environmental information at low temperatures [14].

Transcriptome refers to the sum of all RNAs transcribed by specific cells or tissues at a certain functional state or developmental stage, RNA sequencing (RNA-seq) is a method of transcriptome analysis using deep sequencing techniques that can be used to identify important physiological pathways in organisms that respond to a variety of conditions, and it is widely used to monitor alterations in the transcriptome of targeted tissues, including new transcripts, splice junction and gene annotations [15,16]. High-throughput transcriptomics has revolutionized the field of transcriptome research by providing powerful screening tools and has been widely used in molecular, genetic, and other research fields [17]. Simon A. Wentworth analyzed the transcriptome sequences information of the *Pimephales promelas* (Cyprinidae) in the low-temperature group (5 °C) and high-temperature group (22 °C) during 30 days of culture using high-throughput sequencing and obtained that the expression of 194 specific transcripts was changed. It was found that lower temperature alters the immune system of freshwater teleost fish, leading to genome-wide upregulation of innate immunity and downregulation of adaptive immunity [18]. Wei Zhang found 2534 genes differentially expressed in the liver and 1622 genes differentially expressed in the brain by transcriptomic analysis of the liver and brain in grass carp under high-temperature stress; according to the analysis of the KEGG results, significant differences in the expression of genes involved in metabolic and immune pathways, such as the cAMP signaling pathway, apoptosis, calcium signaling pathway, lipid metabolism, protein processing in the endoplasmic reticulum and the peroxisome proliferator-activated receptor signaling pathway [19]. Since specific responses of different tissues were different under heat stress, many heat stress-related studies used different tissues of fish, such as the liver, muscle, heart, kidney, brain, and gill. In which skeletal muscle exhibited coordinated inhibition of genes constituting sarcomere structure (e.g., actin, myosins, tropomyosin) and those involved in muscle contraction (e.g., parvalbumins and troponins), which can indicate the structural remodeling of muscle tissue in cold [20]. Therefore, muscle tissues were used to compare and analyze the transcriptome information of acrossocheilus in response to high and low temperatures.

In this study, the transcriptome information of the muscle of *A. fasciatus* was analyzed by Illumina sequencing technology, and the genes differentially expressed under long-term stress at different temperatures and their related metabolic pathways were screened. The results of this study can provide fundamental data to elucidate the mechanisms of temperature acclimation in *A. fasciatus* and can expand our understanding of the challenges of environmental adaptation in *A. fasciatus*.

## 2. Results

### 2.1. Evaluation of Transcriptome Data

In this experiment, muscle tissues from the treatment groups 12 °C (T12), 16 °C (T16), 24 °C (T24), and 28 °C (T28) and control groups 20 °C (T20) were subjected to transcriptome sequencing analysis. Through the Illumina Novaseq6000 platform, a total of 114.64 Gb of clean data were obtained from the experimental samples, of which the clean data of each sample reached 6.33 Gb or above, an average GC content of 47.78%, Q20 (percentage of bases with base recognition accuracy greater than 99%) above 97.60% and Q30 (percentage of bases with base recognition accuracy greater than 99.9%) above 93.34% (Table 1) ensuring subsequent assembly.

According to the results of Trinity2.8.5 produced 242,814 unigenes, with an average length of 940 bp and an N50 of 1311 bp (Table 2). The length distribution of the genes is shown in Figure 1. The clean reads from each sample were compared back to the reference sequence, and the alignment rate was 80.82 ± 1.10% (Table 3). The results of the unigenes assembled in this experiment showed that the sequencing quality and other splicing results were accurate and reliable.

### 2.2. Functional Annotation of Unigene

The sequence comparison of all Unigenes in NR, NT, Swissprot, GO, KEGG, and KOG databases showed that 224,722 (92.55%) Unigenes were annotated in the NT database, followed by NR, Swissprot, GO, and KOG database, accounting for 60,536 (24.93%), 34,157 (14.07%), 26,807 (11.04%), 25,657 (10.57%), and only 16,660 (6.86%) Unigenes were annotated by the KEGG database (Table 4).

### 2.3. Differential Gene Expression and Clustering Analysis

When detecting differentially expressed genes (DEGs) between the T20 and T12 groups, the results show that 9202 genes were differentially expressed, of which 4654 and 4548 genes were considered to be up- and down-regulated. Additionally, 4959 genes were shown to be differentially expressed between the T20 and T16 groups, of which 2983 and 1976 genes were considered to be up-and down-regulated. Furthermore, 133 genes were shown to be differentially expressed between the T20 and T24 groups, of which 17 and 116 genes were considered to be up-and down-regulated. Moreover, 878 genes were shown to be differentially expressed between the T20 and T28 groups, of which 343 and 535 genes were considered to be up-and down-regulated (Table 5), and the Venn diagram in Figure 2 showed the statistics of the number of DEGs in the four temperature treatment groups and the control group, there were four identical DEGs in five temperature groups. In order to visualize the overall distribution of all genes in the muscles of the four different temperature experimental groups and the control group of *A. fasciatus*, heat maps (Appendix A) and volcano plots (Appendix A) of the differential genes of the samples were drawn in accordance with the predetermined set threshold values.

### 2.4. KEGG Pathway Enrichment Analyses

To explore the functions of genes related to temperature fluctuations in *A. fasciatus*, the top 20 KEGG enrichment pathways for the four different temperature treatments compared to the control are shown in Figure 3, with the horizontal axis showing the degree of enrichment (indicated by the *p*-value, the smaller the value, the higher the degree of enrichment) and the vertical axis shows the enriched KEGG pathway Terms, the size of the dots indicating the number of differential genes contained in the pathway. The results showed that DEGs in the T12 and T16 groups were mainly enriched in tight junction (ko04530), PI3K-Akt signaling pathway (ko04151), and focal adhesion (ko04510). In the T24 group, DEGs were mainly enriched in proximal tubule bicarbonate reclamation (ko04964), protein digestion and absorption (ko04974), and bile secretion (ko04976). In the T28 group, DEGs were mainly enriched in tight junction (ko04530), glycolysis/gluconeogenesis (ko00010), and biosynthesis of amino acids (ko01230).

### 2.5. RT-qPCR Validation of Transcriptome Data

To further evaluate our DEGs library, eight genes were randomly selected for qRT-PCR analysis. The relative expression levels of the eight genes in the four treatment groups compared with the control group are shown in Figure 4. The results showed that the variation trends were consistent with the transcriptome sequencing analysis, which indicated the reliability of the sequencing results.

## 3. Discussion

Temperature changes affect almost all biochemical and physiological activities of fish [21]. The adaptive capacity of organisms corresponds directly to the changes in their living environment and to their long-term evolution [22], and in order to cope with the negative effects of temperature changes, fish have developed various regulatory mechanisms to adapt their organisms to environmental changes. It has been shown that temperature changes affect the metabolic rate [23], stress response [24], and immune regulation [25] of fish. Therefore, in this study, we used muscle tissues of *A. fasciatus* as experimental material to examine the variations in transcript levels under various temperature stresses using high-throughput sequencing technology, which to some certain extent, enhanced the transcriptome database of *A. fasciatus*.

### 3.1. Process and Pathway of DEG under Temperature Stress

To determine the functional changes of potentially related genes for temperature fluctuation in *A. fasciatus*, we performed KEGG functional analysis of DEGs in four temperature treatment groups, which provided valuable information for studying specific cellular processes, environmental information processing and metabolic organismal system in *A. fasciatus* under temperature fluctuation. We found significant differences in the pathways of protein digestion and absorption, which was similar to the results of Xiang Zhao in his temperature adaptation experiments on yellow drum [26]. In the KEGG results of the high-temperature treatment group and the control group, we found that the HIF-1 signaling pathway was found to be closely related to the physiological function of the organism, and in the KEGG results of the low-temperature treatment group and the control group, we found that the PI3K-Akt signaling pathway was found to be similarly related to the physiological function of the organism.

### 3.2. Effects of Temperature Stress on Fish Metabolism

Changes in the metabolic processes of the organism are one of the most important responses to temperature stress [27]. Our findings demonstrated that the four treatment groups and the control group had significantly different protein digestion and absorption pathways.

In the protein digestion and absorption pathway, transmembrane ATPases (ATPases) are charge-transfer complexes that catalyze ATP synthesis by allowing ions to cross the membrane [28]. The ATPase harnesses the chemical potential energy of ATP by performing mechanical work; the ATPase maintains the sodium-potassium exchanger (Na+/K+ATPase) of the cell membrane potential by importing metabolites essential for cellular metabolism and exporting toxins, wastes, and solutes that may obstruct cellular processes [29]. The potassium channel subfamily K member 5 (KCNK5), which carries out a number of tasks, including controlling cell volume, stabilizing and excitability of the membrane potential, and regulating hormone or ion secretion, is crucial for maintaining normal plasma potassium levels [30]. The endoglucanase celA is essential for the protein that plays an important role in protein synthesis.

In the study, we found that temperature variations significantly affected CELA, KCNK5, and ATPases (Appendix A). As shown in Figure 5, CELA, KCNK5, and ATPase genes were significantly up-regulated in *A. fasciatus* under low temperature, indicating that cellular metabolism was accelerated and the energy demand of the organism was increased, which promoted the protein digestion and absorption pathway [31] and ensured the protein levels of the organism, and to a certain extent ensured the normal nutrition. At 24 °C, KCNK5 was significantly up-regulated, and ATPases and CELA were significantly down-regulated, indicating accelerated metabolism in the organism. At 28 °C, KCNK5 and ATPases were not significantly expressed, while CELA was significantly down-regulated, indicating relatively slower cellular metabolism in the high-temperature environment. The interaction of these genes may protect the fish from potential damage from temperature changes.

### 3.3. Effects of High-Temperature Stress on Fish Stress and Regulation of Major Pathways

Our results showed that the pathway of HIF-1 signaling pathway differed significantly at high temperatures. Stress, including heat stress, oxidative stress, or inflammation [32]. Hypoxia inducible factor 1 (HIF-1) and its signaling pathway play an important role in the regulation of high-temperature environments [33], and the expression of the HIF-1 gene can promote its downstream hypoxia-responsive genes, which are crucial for the body’s response to low oxygen levels or hypoxia [34]. Under hypoxic response conditions in fish, HIF-1 regulates the transcription of hundreds of genes in a cell-specific expression and acts as a major regulator of many hypoxia-inducible genes under hypoxic conditions [35]. Adaptive responses to O_2_ deprivation are mediated by proteins that are expressed by HIF-1 target genes [36]. HIF-1 activity is regulated by the prolyl hydroxylase (PHD) family in an oxygen-dependent manner [37].

In the experiment, we found that high temperature elevated the expression of HIF-1, PHD, and TF in muscle tissue (Appendix A). As shown in Figure 6, when the organism sensed the hypoxia signal, HIF-1 expression was significantly up-regulated, and PHD expression was up-regulated, thus increasing the adaptability to high temperature and hypoxia response. The expression of the serum transferrin (TF) gene in the pathway was down-regulated, thus affecting the iron metabolic pathway, and the down-regulation of TF gene expression also revealed that the prolonged exposure of the organism to high temperature caused damage to the liver [38], which had an impact on the *A. fasciatus’s* overall health. In our experiment, even though oxygen saturation had been reached in the water environment, the fish organism was still in a hypoxic state, indicating that the high-temperature stress process increased the expression of HIF-1 and PHD, to some extent avoiding the potential damage of high temperature to the *A. fasciatus.*

### 3.4. Effects of Low-Temperature Stress on Fish Immunity and Regulation of Major Pathways

Our results showed that the pathway of PI3K-Akt signaling pathway differed significantly at low temperatures. The effect of temperature stress on fish immune systems has become a hot topic of research, and the signal transduction mechanisms related to immune system activation remain to be elucidated. Low-temperature stress may damage the immune defense system and increase its susceptibility to pathogenic bacteria [39].

In the experiment, we found that low-temperature stress increased the expressions of AKT, FOXO3a, BAD, GSK-3, and AMPK in muscle tissue (Appendix A). As shown in Figure 7, AKT phosphorylation enhanced its interaction with the downstream effector protein FOXO3a, thereby inhibiting autophagic activity [40]. When the heart is hypoxic and ischemic, the volume load and pressure load increase, or the pro-inflammatory factor increase, and oxygen radical production increase, causing oxidative stress [41]. AKT also inhibited the Bcl-2/Bel-XL associated death promoter gene (BAD), causing significant down-regulated of the BAD and preventing anti-apoptotic effects in the organism [42]. Another mechanism of AKT in protecting early ischemic myocardium is to ensure the minimum energy supply during myocardial hypoxia by inhibiting GSK-3 and promoting glycolysis [43]. Activation of the PT3K/AKT signaling pathway could inhibit GSK-3 downstream, thereby preventing the opening of mitochondrial permeability transition pore (mPTP) and protecting myocardial muscle by inhibiting apoptosis or promoting proliferation [44]. GSK-3 was significantly up-regulated in the pathway data, indicating that prolonged hypothermic stress may be harmful to myocardial survival and cardiac function. The PI3K-Akt signaling pathway, a key intracellular sensor and regulator of energy homeostasis that can be activated by ATP depletion or an increased AMP/ATP ratio, significantly up-regulated AMPK expression [45]. 

## 4. Materials and Methods

### 4.1. Experimental Animals

The present study was conducted under the permission and supervision of the Institutional Animal Care and Use Committee of Zhejiang Ocean University. The approval number is 20210207.

*A. fasciatus* were obtained from the breeding base in Ta-pieh Mountains, Anhui, China. The fish were divided in the first stage of the experiment and were temporarily cultured in a tank for 2 weeks at a water temperature of 17 ± 1 °C; 450 healthy individuals with symmetrical body shapes, no obvious mechanical or artificial damage on the body surface, and similar and active individuals were randomly selected as the experimental materials. The average body length was 0.57 ± 0.16 cm, and the average body weight was 4.08 ± 0.40 g. During the experiment, fish were fed at 9:00 am and 18:00 pm (feed weighed at 2% of body weight), fish growth was checked, dead and injured fish were retrieved, the water temperature was measured with a thermometer and recorded, fish excrement and other foreign objects were removed from the tanks, and ventilation and a quiet and comfortable environment were maintained at all times.

### 4.2. Temperature Stress Experiment

Five temperature groups were designed for this experiment, and experimental fish were cultured for 60 days at different temperatures of 12 °C (T12), 16 °C (T16), 20 °C (control, T20), 24 °C (T24) and 28 °C (T28) in circulating water, 30 fish per tank, and each group was performed in triplicate. The experimental culture tank of each temperature group was placed in 5 temperature boxes (Shanghai Yiheng LRH-150 biochemical incubator, Shanghai, China). Each temperature group was cultured at a rising and cooling rate of 1 °C/24 h until the temperature reached the experimental water temperature adaptation for a week and maintained for 60 days. The feeding was stopped 1 d before sampling, and 10 fish were randomly selected from each tank, with a total of 30 fish in each treatment group, dissected on ice, take the back muscles of the fish and the muscle tissues were immediately snap frozen in liquid nitrogen and then stored at −80 °C in the freezer.

### 4.3. Total RNA Extraction and Illumina Sequencing

There were 3 parallel samples per temperature group, using the RNA extraction kit (TaKaRa, Code: No. 9767) according to the manufacturer’s instructions, RNA was extracted from each group of muscle tissue, and each group of RNA was mixed in equal proportions for a total of 15 samples. The Agilent 2100 bioanalyzer (Agilent Technologies, Santa Clara, CA, USA) was used to measure each RNA concentration; RNA 28S/18S ≥ 1.5 and RNA integrity number >7.0 were selected for further analysis. RNA availability and integrity by 1% agarose gel electrophoresis, and then high throughput sequencing via Illumina Novaseq6000 (Berry Genomics Co., Ltd., Beijing, China).

### 4.4. Sequencing Data Quality Control, Assembly, and Gene Function Annotation

The sequenced raw reads were filtered to remove spliced, duplicated, low-quality sequences, resulting in clean reads. The sequences were assembled using Trinity 2.8.5 to obtain transcripts, and the Unigene of the corresponding species was obtained based on the component information, and the Unigene was compared with NR, NT, Swissprot, Gene Ontology (GO), Kyoto Encyclopedia of Genes and Genomes (KEGG), and Eukaryotic Ortholog Groups (KOG)/Clusters of Orthologous Groups of proteins (COG) databases respectively to obtain Unigene annotation information.

### 4.5. Differentially Expressed Gene Analysis, Functional and Pathway Enrichment

Using the transcriptome assembled by Trinity as the reference sequence. The clean reads of each sample were compared back to the reference sequence using bowtie2 2.3.2 software, the RSEM counted the results of the bowtie2 comparison and further obtained the number of read counts of each sample compared to each Unigene and converted them to the number of bases per thousand (FPKM) compared to the transcript per million compared fragments. The edgeR 3.3.3 software was used to analyze the differential expression of genes in each sample and calculate the *p*-value and padj values of DEGs; padj is the corrected *p*-value. The smaller the padj value, the more significant the gene expression difference. To control the false positive rate, the padj value combined with Fold Change was required to screen for differential genes, and the screening conditions are as follows: padj < 0.05 and |log2FoldChange| > 1, and cluster analysis was performed on the DEGs obtained from different temperature culture. To further understand the function of DEGs at different temperatures, KEGG enrichment analysis using KOBAS 3.0 software [46] was used to classify DEGs in different temperature groups, and the significance of differential gene enrichment in each pathway entry was calculated by a hypergeometric distribution test. Each functional module had a *p*-value, and the false discovery rate (FDR) correction for the *p*-value, the function of FDR ≤ 0.05, was set as a significant enrichment.

### 4.6. Quantitative Real-Time PCR

We used PrimeScript^TM^ RT Master Mix kit (TaKaRa, Kusatsu, Japan, Code: RR036A) to reverse transcribe the total RNA to obtain the cDNA template used in the qRT-PCR experiment. According to the results of the standard curve, samples were diluted 10-fold in nuclease-free water as templates for qRT-PCR. According to the manufacturer’s instructions, qRT-PCR experiments using TB Green^®^ Premix Ex Taq^TM^ II kit (TaKaRa, Code: RR820A). A 20 µL reaction system was amplified, including 10 µL of TB Green Premix Ex TaqII (2×), 2 µL of cDNA template, 0.4 µL of ROX Reference Dye (50×), 0.8µL of PCR Forward Primer (10 µM), 0.8µL of PCR Reverse Primer (10 µM) and 6 µL of RNase-free water. The amplification processes consisted of a holding stage of 30 s at 95 °C, followed by 40 cycles of 5 s at 95 °C and 30 s at 60 °C. Three parallel experiments on each cDNA template were performed to reduce the error of the experimental results. Primer Premier 6 [47] software was used to design gene-specific primers with β-actin as an internal reference gene (Table 6) and sent to Sunya Biotechnology Co., Ltd., Nanjing, China for synthesis. The relative expression of each of the 8 genes was analyzed using the comparative cycle threshold (2^−ΔΔCT^) method (ΔCT = CTtarget gene − CTreference gene, ΔΔCT = ΔCTtreatment −ΔCTcontrol). SPSS 19.0 [48] was used for statistical analysis, and the one-way ANOVA and *t*-test were used to determine the significance of differences. *p*-values < 0.05 were considered statistically significant.

## 5. Conclusions

Under the condition of temperature change, the physiological activities of the *A. fasciatus* have changed. Muscle transcriptome analysis of the *A. fasciatus* at five different temperatures showed that the protein digestion and absorption pathways were mainly regulated at temperature changes, the immune system was mainly regulated at low temperatures, and the hypoxia system was mainly regulated at high temperatures. We found that the physiological metabolism of muscle was significantly affected during the change in temperature. The findings of this work provided basic muscle transcriptome data for the future study of the temperature adaptation of Orientalis obscurus.

## Figures and Tables

**Figure 1 ijms-24-11622-f001:**
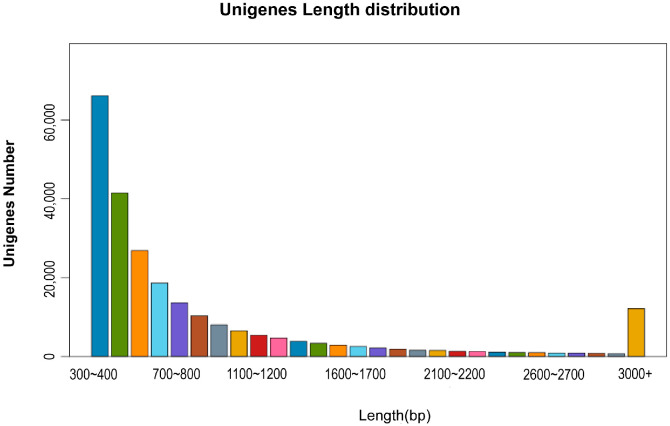
Unigene length is one of the indicators reflecting assembly quality. The statistical results of the Unigene length distribution are as follows.

**Figure 2 ijms-24-11622-f002:**
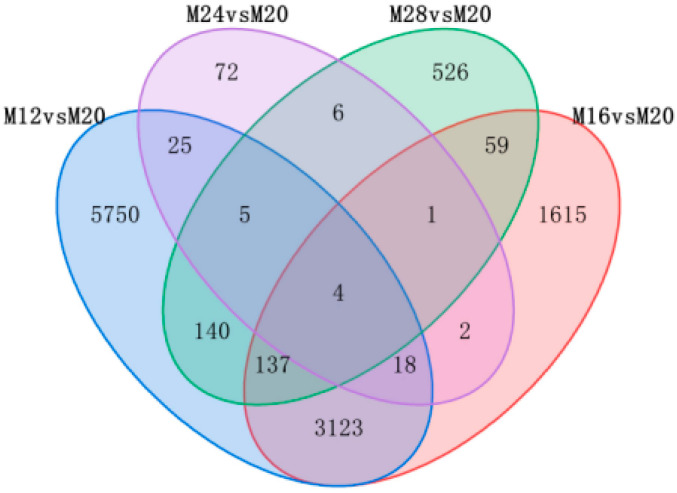
The Venn diagram showed DEGs between different temperature groups and control groups.

**Figure 3 ijms-24-11622-f003:**
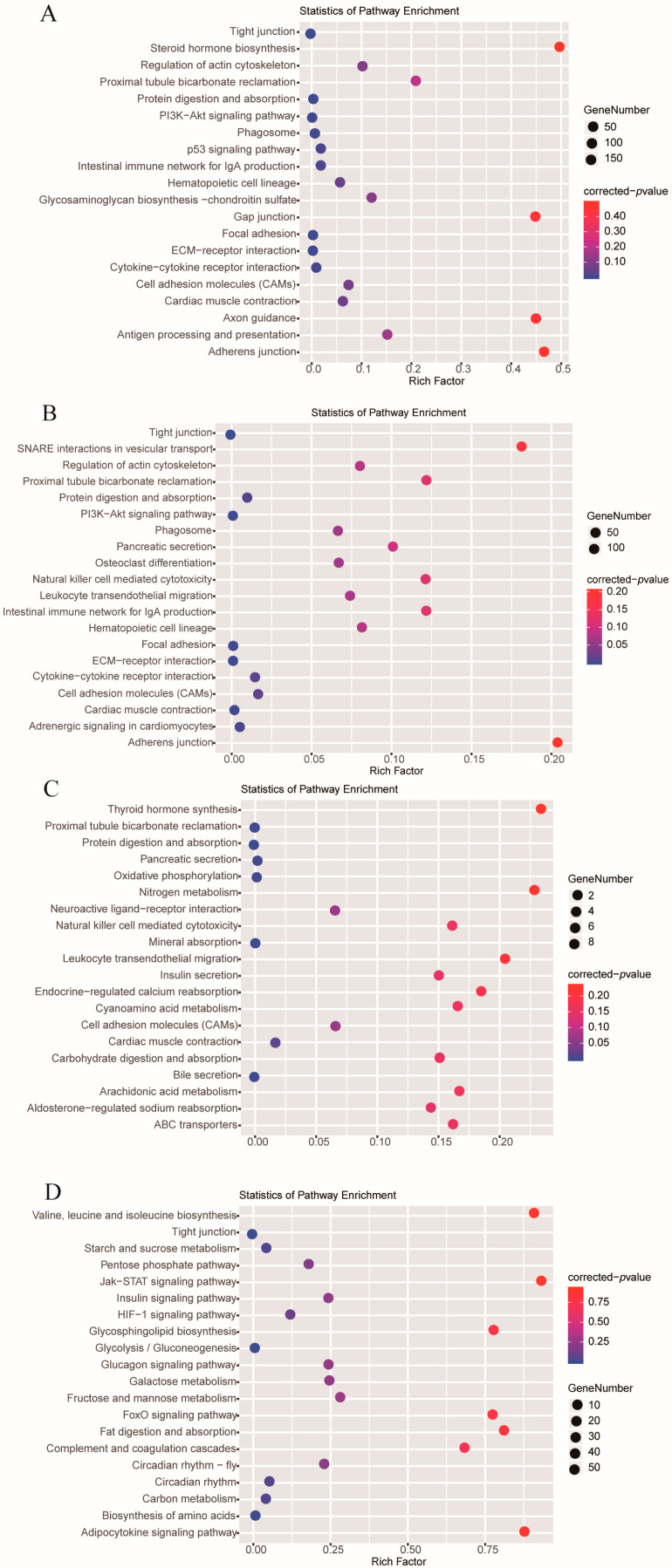
The top 20 pathways of KEGG enrichment, the size of the bubble indicated the number of genes enriched in a KEGG pathway, and the color represented the Q value. (**A**) T12 vs. T20; (**B**) T16 vs. T20; (**C**) T24 vs. T20; (**D**) T28 vs. T20.

**Figure 4 ijms-24-11622-f004:**
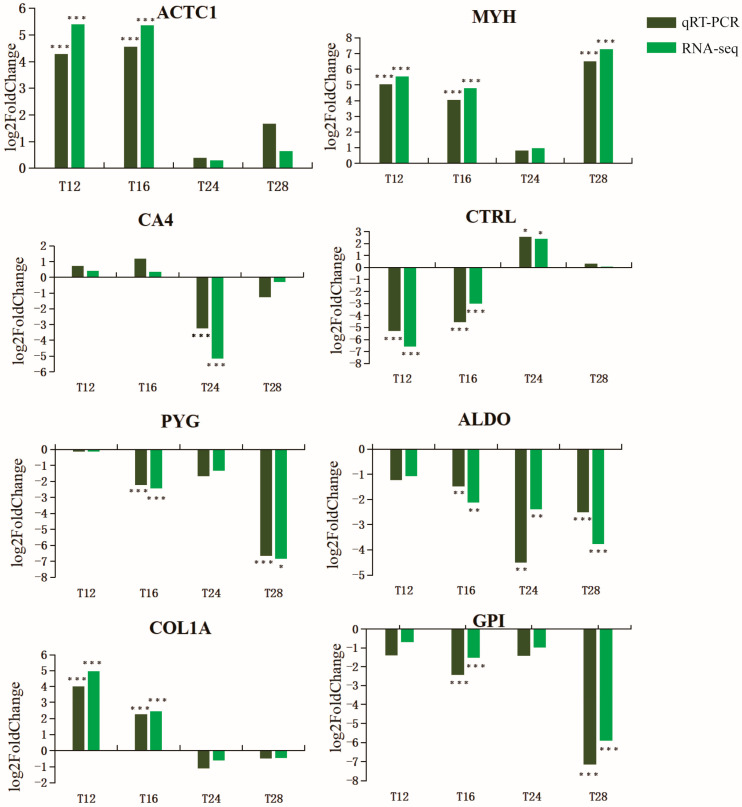
RT-qPC verifies RNA-Seq results to ensure the accuracy of transcriptome data. (* *p* < 0.05, ** *p* < 0.01, *** *p* < 0.001).

**Figure 5 ijms-24-11622-f005:**
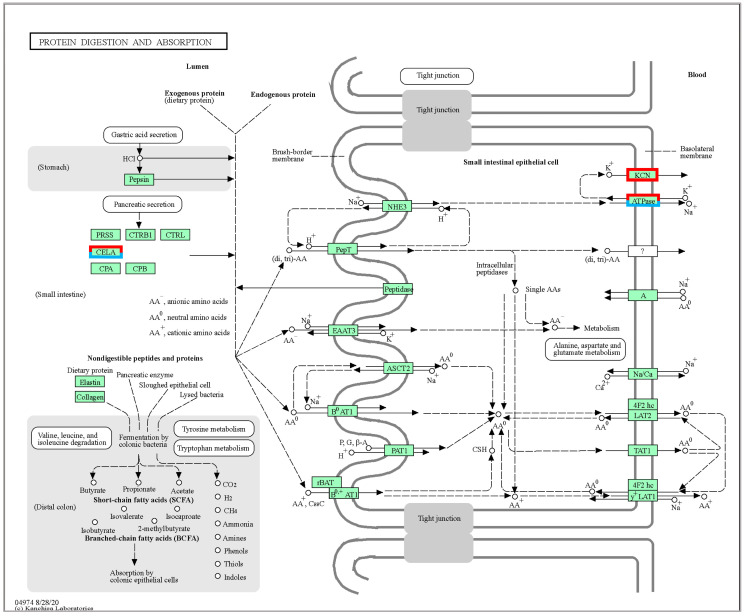
Significantly DEGs identified by KEGG in protein digestion and absorption pathways (ko04974). Red boxes indicate genes that are significantly up-regulated, and both red and blue boxes indicate genes that are both up-regulated and down-regulated.

**Figure 6 ijms-24-11622-f006:**
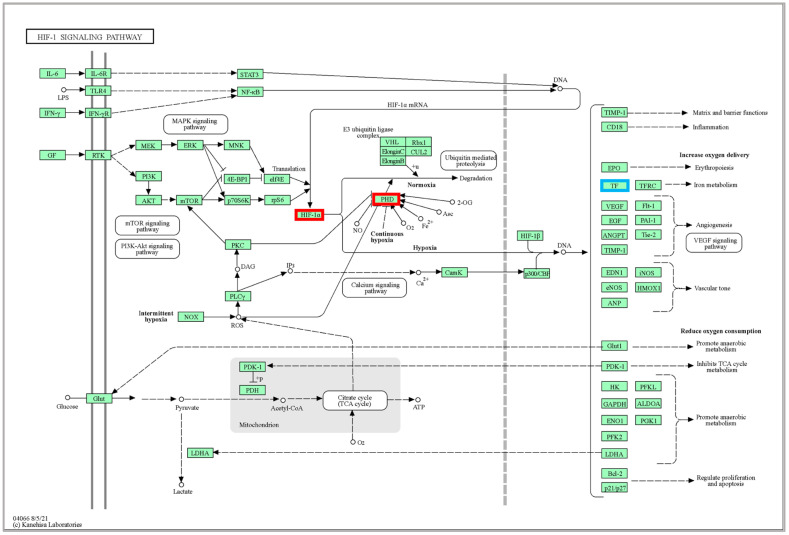
Significantly DEGs identified by KEGG in the HIF-1 signaling pathway (ko04066). The red box represents genes that are significantly up-regulated, and the blue box represents genes that are significantly down-regulated.

**Figure 7 ijms-24-11622-f007:**
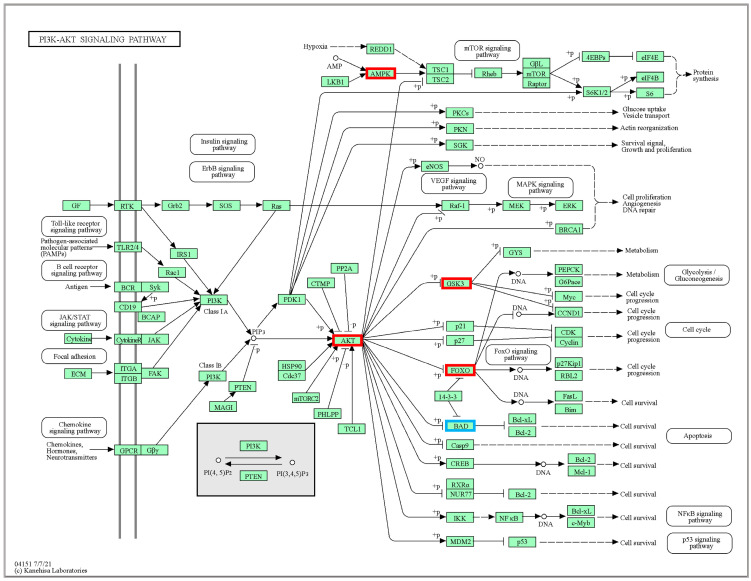
Significantly DEGs identified by KEGG in PI3K-Akt signaling pathway (ko04151). The red box represents genes that are significantly up-regulated, and the blue box represents genes that are significantly down-regulated.

**Table 1 ijms-24-11622-t001:** Quality analysis of filtered sequences and summary of reads of the samples retrieved from Illumina sequencing.

Sample Name	Clean Reads	Clean Bases	Clean N (%)	Clean GC Ratio (%)	Clean Q20 Ratio (%)	Clean Q30 Ratio (%)
T12-1	24,316,773	7,295,031,900	0	46.73	97.86	94.10
T12-2	25,410,326	7,623,097,800	0	47.24	97.86	94.11
T12-3	24,898,312	7,469,493,600	0	46.87	97.98	94.48
T16-1	32,794,432	9,838,329,600	0	46.70	97.83	94.22
T16-2	23,386,950	7,016,085,000	0	46.77	97.77	94.20
T16-3	21,114,837	6,334,451,100	0	47.95	97.96	94.38
T20-1	27,319,993	8,195,997,900	0	47.89	97.97	94.48
T20-2	24,386,348	7,315,904,400	0	48.50	98.04	94.47
T20-3	30,775,172	9,232,551,600	0	48.19	98.14	94.67
T24-1	24,653,640	7,396,092,000	0	48.31	97.82	94.15
T24-2	25,124,451	7,537,335,300	0	48.24	98.12	94.73
T24-3	24,181,637	7,254,491,100	0	48.76	97.70	93.92
T28-1	26,557,598	7,967,279,400	0	48.81	97.72	93.61
T28-2	22,807,447	6,842,234,100	0	47.76	97.60	93.34
T28-3	24,392,064	7,317,619,200	0	48.05	97.68	93.61

**Table 2 ijms-24-11622-t002:** Unigene length, N50 length, and Mean length distribution.

Unigene Total Number	Unigen Total Length/bp	N50 Length/bp	Mean Length/bp
242,814	22,838,2225	1311	940.5644856

Note: N50: Sort the spliced transcripts in order of length from longest to shortest and accumulate the lengths of the transcripts to no less than 50% of the total length of the spliced transcripts.

**Table 3 ijms-24-11622-t003:** Statistical summary of RNA-seq read, and reference sequence comparison results for mapping.

Sample Name	Total Reads	Mapped Reads (Ratio%)
T12-1	48,633,546	38,867,266 (79.92%)
T12-2	50,820,652	41,480,416 (81.62%)
T12-3	49,796,624	40,270,714 (80.87%)
T16-1	65,588,864	52,932,638 (80.70%)
T16-2	46,773,900	37,384,300 (79.93%)
T16-3	42,229,674	34,927,104 (82.71%)
T20-1	54,639,986	43,569,030 (79.74%)
T20-2	48,772,696	40,317,360 (82.66%)
T20-3	61,550,344	50,379,846 (81.85%)
T24-1	49,307,280	39,098,918 (79.30%)
T24-2	50,248,902	40,699,492 (81.00%)
T24-3	48,363,274	38,236,040 (79.06%)
T28-1	53,115,196	43,490,240 (81.88%)
T28-2	45,614,894	36,763,338 (80.60%)
T28-3	48,784,128	39,258,654 (80.47%)

**Table 4 ijms-24-11622-t004:** Statistical table of functional annotations of DEGs and the number of annotated differentially expressed genes (DEGs). When Unigene annotates three or more different databases at the same time, the accuracy of Unigene’s annotation is demonstrated.

Anno_Database	Annotated_Number	Percentage/%
GO_Annotation	26,807	11.04%
KEGG_Annotation	16,660	6.86%
KOG_Annotation	25,657	10.57%
NR_Annotation	60,536	24.93%
NT_Annotation	224,722	92.55%
Swissprot_Annotation	34,157	14.07%
Total unigenes	242,814	100.00%

**Table 5 ijms-24-11622-t005:** Statistical table of DEGd number based on Illumina sequencing, including total gene number and up-down-regulated gene number.

Type	Total	Up	Down
T12 vs. T20	9202	4654	4548
T16 vs. T20	4959	2983	1976
T24 vs. T20	133	17	116
T28 vs. T20	878	343	535

**Table 6 ijms-24-11622-t006:** The genes and gene-specific primers used for qRT-PCR.

Gene	Forward Primer Sequence	Reverse Primer Sequence
*MYH*	CTGTCAAGAGCATCAATGAC	CGACCTTCACTCTGGGGTAG
*GPI*	AGCCGCATCATGTGATTCATC	TCGTATGGTTCCTGCTGACTT
*ALDO*	GGTGATTAGTAGGCATGGTTGG	CATCTCCAGAACAACAGCAAGG
*CTRL*	CCACTGCTCTGTTGCGGTCA	AGGCTGGATGCTGATGGTTGAT
*CA4*	CTGTTCCTTGCTGAGCGGTATG	CCTGGCACCAATGTAACACTGT
*COL1A*	GCAGCAACACAGCCTTCTTCA	GGTTCGGCGAGACCATCAATG
*PYG*	AAGAGCCTAACAAGCAGTGGA	TGGGAAGAGTTGAGCAGAACA
*ACTC1*	ATGTGCGACGAGGAAGAGACC	CAGTTGGTGATGATGCCATGCT
*β-actin*	CCCAGAATCCTATTGTTACCC	CCTCGCATACATAGTGCCATT

## Data Availability

Not applicable.

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
