# Peer review of "Transcriptional Modulation Reveals Physiological Responses to Temperature Adaptation in Acrossocheilus fasciatus"

_ijms, 2023, doi:10.3390/ijms241411622_

Round 1

Reviewer 1 Report

This study investigated the effect of various water temperatures on gene expression profiles in the muscle of A. fasciatus. While the data are interesting and contribute to the current understanding of the stress responses in A. fasciatus, the manuscript would benefit from revisions and clarifications of many issues raised below. It is good to see that the RNA-seq data show consistency with the qPCR data. However, much of the RNA-seq and qPCR protocol information needs to be provided, see comments below. Also, it is recommended that the author needs to thoroughly check the English grammar in the manuscript.

Abstract

L12-14: suggest specifying which of those pathways were in the elevated temperature groups and which were from cold groups.

Introduction

L27 – please check the English grammar here and throughout the manuscript.

L35 – Please clarify whether it is air or water temperature.

L37-40 – repeated information. Suggesting partially deleting these sentences.

L49-51 – please check the grammar.

L65, L71 – here and throughout the manuscript, check reference style

Materials and Methods

L311 – it would be helpful to provide the ethics approval ID.

L314 – what was the water temperature then?

L317 –  0.57 ± 0.16 cm, is this mean±SD? Please clarify here the below.

L335 – should be a freezer.

L341 – If possible, please provide the mean±SD of RNA integrity number (RIN) and 28/18S ratio used for the downstream RNA-seq analysis. This is to make sure that decent quality of RNAs was used.

L343- please provide the kit manufacturer and system names for cDNA library construction. Also please disclose further the sequencing information, e.g., length, pair-end reads?

L366-367 confusing sentences.

L370 – Did the author considering analysing the up- and down-DEG in each comparison separately? It seems that the author combined the up- and down- genes in each comparison group for analysis. Please provide the rationale of doing this.

L372 – For KEGG what was the adjP threshold for significant enrichment? Please clarify.

L374 – For the qPCR – was the target gene amplification efficiency being tested? Please confirm that the guideline has been followed.  Bustin SA, Benes V, Garson JA, Hellemans J, Huggett J, Kubista M, Mueller R, Nolan T, Pfaffl MW, Shipley GL, Vandesompele J. The MIQE Guidelines: Minimum Information for Publication of Quantitative Real-Time PCR Experiments.

Results

L97 – please use mean±SD or mean±SEM where possible.

L128 – “with a total of 4 DEGs for the temperature..” confusing sentence.

Figure 4 – in the figures A and B, the comparisons are M20vsM12 and M20vsM16, whereas C and D are compared to relative ‘control (M20)’. Please make sure the comparison direction is consistent, and the up- and down-regulation relative to control are expressed correctly in the figure.

L145, When reporting specific GO terms, eg., tight junction, please include the pathway ID/identifier (GO:XXXX) with adjP values. Also, did the author considering analysing the up- and down-regulated genes separately in each comparison?

Discussion

L233, here and throughout the manuscript, please avoid using very short sentences as an individual paragraph. Suggesting merging into the next one.

L268-278, The background information needs to be shortened. Suggesting shortening this section and merging it with the next paragraph.

English grammar needs to be checked.

Author Response

Response to Reviewer 1 Comments

Point 1: I Abstract

L12-14: suggest specifying which of those pathways were in the elevated temperature groups and which were from cold groups.

Response 1: It has been revised in the abstract.

‘Compared with 20 ℃, 9202, 4959 differentially expressed genes (DEGs) were discovered in low temperatures groups(12 ℃, 16 ℃), whereas 133 and 878 DEGs were discovered in high temperatures groups(24 ℃, 28 ℃), respectively. According to the KEGG functional enrichment analysis revealed that DEGs were primarily enriched in tight junction, PI3 K-Akt signaling pathway and protein digestion and absorption in low temperature groups, and mainly enriched in proximal tubule bicarbonate reclamation, protein digestion and absorption and HIF-1 signaling pathway in high temperature groups.’.

Point 2: Introduction

L27 – please check the English grammar here and throughout the manuscript.

Response 2: The English grammar has been checked in the attachment.

Point 3: Introduction

L35 – Please clarify whether it is air or water temperature.

Response 3: After confirmation, ‘It has been demonstrated that the A. fasciatus can survive in a water temperature range of approximately 12–30℃’.

Point 4: Introduction

L37-40 – repeated information. Suggesting partially deleting these sentences.

Response 4: After your suggestion, this section is simplified and merged.

Point 5: Introduction

L49-51 – please check the grammar.

Response 5: The revised sentence is ’Fish are capable of adapting to environmental temperature by modifying their physiological activity within a specific temperature range, which has become a popular research topic.’

Point6: Introduction

L65, L71 – here and throughout the manuscript, check reference style

Response 6: These two have been amended.

‘Simon A. Wentworth analyzed the transcriptome sequences......’

‘Wei Zhang found 2534 genes differentially expressed in liver and ......’

Point 7: Materials and Methods

L311 – it would be helpful to provide the ethics approval ID.

Response 7:It has been proposed in the article. ’The approval number is 20210207’.

Point 8: Materials and Methods

L314 – what was the water temperature then?

Response 8: The water temperature is 17±1°C.

 SO it has been modified in the article ‘at a water temperature of 17±1 °C’.

Point 9: Materials and Methods

L317 –  0.57 ± 0.16 cm, is this mean±SD? Please clarify here the below.

Response 9: Yes, it is the mean±SD.

Point 10: Materials and Methods

L335 – should be a freezer.

Response 10: Yes, SO it has been modified in the article ‘......in the freezer’.

Point 11: Materials and Methods

L341 – If possible, please provide the mean±SD of RNA integrity number (RIN) and 28/18S ratio used for the downstream RNA-seq analysis. This is to make sure that decent quality of RNAs was used.

Response 11:This has been added in the article ’RNA 28S/18S≥1.5 and RNA integrity number >7.0 were selected for further analysis. ’

Point 12: Materials and Methods

L343- please provide the kit manufacturer and system names for cDNA library construction. Also please disclose further the sequencing information, e.g., length, pair-end reads?

Response 12:This has been added in the article ‘...illumina Novaseq6000 (Berry Genomics Co., Ltd, in China).

Point 13: Materials and Methods

L366-367 confusing sentences.

Response 13: Yes,so we delete it.

Point 14: Materials and Methods

L370 – Did the author considering analysing the up- and down-DEG in each comparison separately? It seems that the author combined the up- and down- genes in each comparison group for analysis. Please provide the rationale of doing this.

Response 14: We not only analyzed the combination of the up- and down- genes in each comparison group in 3.3, 3.4, but also analyzed the up- and down-DEG in each comparison separately in 3.2.

Our analysis is based on the experimental results. At low temperature, the genes in the PI3K-Akt signaling pathway changed significantly only at low temperature, at high temperature, the genes in the HIF-1 signaling pathway changed significantly only at high temperature. When the temperature changes, DEGs in the protein digestion and absorption changes in temperature.

Point 15: Materials and Methods

L372 – For KEGG what was the adjP threshold for significant enrichment? Please clarify.

Response 15: It has been proposed in the article. ’Each functional module had a p-value, the false discovery rate (FDR) correction for p-value, the function of FDR ≤ 0.05, was set as a significant enrichment.

Point 16: Materials and Methods

L374 – For the qPCR – was the target gene amplification efficiency being tested? Please confirm that the guideline has been followed.  Bustin SA, Benes V, Garson JA, Hellemans J, Huggett J, Kubista M, Mueller R, Nolan T, Pfaffl MW, Shipley GL, Vandesompele J. The MIQE Guidelines: Minimum Information for Publication of Quantitative Real-Time PCR Experiments.

Response 16:It has been changed in the article ’We used PrimeScriptTM RT Master Mix kit (TaKaRa, Code: RR036A) to reverse transcribe the total RNA to obtain the cDNA template used in the qRT-PCR experiment. According to the results of the standard curve, samples were diluted 10-fold in nuclease-free water as templates for qRT-PCR. According to the manufacturer’s instructions, qRT-PCR experiments using TB Green® Premix Ex TaqTM II kit (TaKaRa, Code: RR820A). A 20 µL reaction system was amplified, including 10 µL of TB Green Premix Ex TaqII (2×), 2 µL of cDNA template, 0.4 µL of ROX Reference Dye (50×), 0.8µL of PCR Forward Primer(10 µM), 0.8µL of PCR Reverse Primer(10 µM) and 6 µL of RNase-free water. The amplification processes consisted of a holding stage of 30 s at 95 ◦C, followed by 40 cycles of 5 s at 95 ◦C and 30 s at 60 ◦C. Three parallel experiments on each cDNA template were performed to reduce the error of the experimental results. Primer Premier 6[47] software was used to design gene-specific primers with β-actin as an internal reference gene (Table 6), and sent to Zhejiang Sunya Biotechnology Co., Ltd, in China for synthesis. The relative expression of each of the 8 genes were analysed using the comparative cycle threshold (2−ΔΔCT) method (ΔCT = CTtarget gene − CTreference gene, ΔΔCT = ΔCTtreatment −ΔCTcontrol). SPSS 19.0[48] was used for statistical analysis and one-way ANOVA and t-test were used to determine the significance of differences. p-values <0.05 were considered statistically significant. ’.

Point 17: Results

L97 – please use mean±SD or mean±SEM where possible.

Response17:It has been proposed in the article. ’and the alignment rate was 80.82±1.10%’.

Point 18: Results

L128 – “with a total of 4 DEGs for the temperature..” confusing sentence.

Figure 4 – in the figures A and B, the comparisons are M20vsM12 and M20vsM16, whereas C and D are compared to relative ‘control (M20)’. Please make sure the comparison direction is consistent, and the up- and down-regulation relative to control are expressed correctly in the figure.

Response 18:It has been proposed in the article. ’ there were 4 identical DEGs in 5 temperature groups.’.

Figure 4 has been modified in the attachment.

Point 19: Results

L145, When reporting specific GO terms, eg., tight junction, please include the pathway ID/identifier (GO:XXXX) with adjP values. Also, did the author considering analysing the up- and down-regulated genes separately in each comparison?

Response 19:It has been proposed in the article. ’The results showed that DEGs in T12 and T16 groups were mainly enriched in tight junction (ko04530), PI3K-Akt signaling pathway (ko04151), and focal adhesion (ko04510). In T24 group, DEGs were mainly enriched in proximal tubule bicarbonate reclamation (ko04964), protein digestion and absorption (ko04974), and bile secretion (ko04976). In the T28 group, DEGs were mainly enriched in tight junction (ko04530), glycolysis/gluconeogenesis (ko00010), and biosynthesis of amino acids (ko01230)’.

We not only analyzed the combination of the up- and down- genes in each comparison group in 3.3, 3.4, but also analyzed the up- and down-DEG in each comparison separately in 3.2.

Our analysis is based on the experimental results. At low temperature, the genes in the PI3K-Akt signaling pathway changed significantly only at low temperature, at high temperature, the genes in the HIF-1 signaling pathway changed significantly only at high temperature. When the temperature changes, DEGs in the protein digestion and absorption changes in temperature.

Point 20: Discussion

L233, here and throughout the manuscript, please avoid using very short sentences as an individual paragraph. Suggesting merging into the next one.

Response 20: The manuscript has been revised and merged.

Point 21: Discussion

L268-278, The background information needs to be shortened. Suggesting shortening this section and merging it with the next paragraph.

Response 21: The contents of 3.4 have been shortened and merged.

Here are some places that we did not list the changes marked in the attachment.

We appreciate for Editors and Reviewers warm work earnestly, and hope that the correction will meet with approval.

Once again, thank you very much for your comments and suggestions.

Reviewer 2 Report

The paper entitled: “Transcriptional Modulation Reveals Physiological Responses to Temperature Adaptation in Acrossocheilus fasciatus” from Zhenzhu Wei, Yi Fang, Wei Shi, Zhangjie Chu, Bo Zhao describes transcriptome analyses of muscle tissues in response to high and low temperature. The data identified several stress related metabolic pathways which are likely involved in temperature acclimation in muscle tissue. The author gave a several information in the introduction and also in the discussion which are redundant. They should focus on the main message for the readers especially in the conclusions focusing on the main findings only. Overall the paper is well written and clear, but it contains too many figures that are not originally produced by the authors.

Below main comments:

1. The authors should better explain why they used only tissues from muscles and why those tissues is more important than others, such as central nerve system or liver, for studying acclimation.

2. The authors should also better clarify which muscle tissue did they used (dissected) and if possible the size and weight. It is important in the future if someone else wish to use a similar approach.

3. How did they dissected the tissues and if they can completely exclude contamination from other tissues.

4. Another important aspects the authors needs to clarify how they selected the temperatures. Why did they have selected those and what was the expected results, if possible. Is this potentially connected to environmental observations in the wild?

5. In Figure 3 the authors made some kind of pairwise comparisons between treatments, why they did not analyses all the treatments together? It would be also necessary to show the comparisons with the controls. All these information should be available in supplementary materials.

6. There are no supplementary materials

7. The paper contains several figures obtained by the KEGG, in which the authors highlighted the DEGs. The authors should cite in the legends that the figure were obtained from KEGG. Moreover, there are too many, I suggest that all those figures should be moved to supp material.

8. All figures and tables legends are too short and not enough informative. The legends should better introduce the figure and table describing when possible the main results. In the text (see below) the authors added information that normally are written in the legend.

9. About the conclusion, the authors should be more careful because their data is only produced using muscle tissues, therefore they should refer to the effects of the different treatments on the muscle tissues only. Other tissues would be also important to define more generic conclusions. The conclusion needs to be carefully rewritten.

Specific comments to the test:

Lines 121 – 128: Can you please define each treatment? The 7 line paragraph is only listing numbers that are not understandable without the figures. Better described in table 5.

Lines 132 – 134: this information fits better to the legends

Lines 235: remove the “_”

It is fine and clear

Author Response

Response to Reviewer 2 Comments

Point 1: 

  1. The authors should better explain why they used only tissues from muscles and why those tissues is more important than others, such as central nerve system or liver, for studying acclimation.

Response 1: Additional information has been added to the manuscript.’Since specific responses of different tissues were different under heat stress, many heat stress-related studies used different tissues of fish such as the liver, muscle, heart, kidney, brain, and gill. In which skeletal muscle exhibited coordinated inhibition of genes constituting sarcomere structure (e.g. actin, myosins, tropomyosin )and those involved in muscle contraction (e.g. parvalbumins and troponins), which can indicate the structural remodeling of muscle tissue in cold.[20]Therefore, muscle tissues were used to compare and analyze the transcriptome information of acrossocheilus in response to high and low temperatures.’

Point 2 

  1. The authors should also better clarify which muscle tissue did they used (dissected) and if possible the size and weight. It is important in the future if someone else wish to use a similar approach.

Response 2: The article has been added’ dissected on ice, take the back muscles of the fish, the muscle tissues were immediately snap frozen in liquid nitrogen and then stored at - 80 °C in the freezer.’

Point 3:

  1. How did they dissected the tissues and if they can completely exclude contamination from other tissues.

Response 3: We put the dissected fish on ice and took the back muscle, which has a low chance of being contaminated with other tissues.

Point 4: 

  1. Another important aspects the authors needs to clarify how they selected the temperatures. Why did they have selected those and what was the expected results, if possible. Is this potentially connected to environmental observations in the wild?

Response 4: 

It is mentioned in our manuscript ‘It has been demonstrated that the A. fasciatus can survive in a water temperature range of approximately 12-30℃[3]’, This is based on the data obtained from reference literature and field environment observation. We initially chose this temperature experiment to study the influence of low and high temperature on Acrossocheilus fasciatus.

Point 5:

  1. In Figure 3 the authors made some kind of pairwise comparisons between treatments, why they did not analyses all the treatments together? It would be also necessary to show the comparisons with the controls. All these information should be available in supplementary materials.

Response 5: Our experimental design is the analysis of different high-temperature and low-temperature groups and control temperature groups. Added to supplemental material.

Point 6: 

  1. There are no supplementary materials.

Response 6: Supplementary materials has been added.

Point 7: 7. The paper contains several figures obtained by the KEGG, in which the authors highlighted the DEGs. The authors should cite in the legends that the figure were obtained from KEGG. Moreover, there are too many, I suggest that all those figures should be moved to supp material.

Response 7: The data from KEGG is already represented in the legend ’The expression of major DEGs of KEGG data in protein digestion and absorption pathways. (*P < 0.05, **P < 0.01, ***P < 0.001)’

The relevant materials has been moved to the supply material.

Point 8: 

  1. All figures and tables legends are too short and not enough informative. The legends should better introduce the figure and table describing when possible the main results. In the text (see below) the authors added information that normally are written in the legend.

Response 8: The figures and tables legends have been revised.

‘ Quality analysis of filtered sequences and summary of reads of the samples retrieved from Illumina sequencing..’

‘ Unigene length、N50 length and Mean length distribution. Note: N50: Sort the spliced transcripts in order of length from longest to shortest and accumulate the lengths of the transcripts to no less than 50% of the total length of the spliced transcripts.’

‘ Statistical summary of RNA-seq read and reference sequence comparison results for mapping.’

‘ Unigene length is one of the indicators reflecting assembly quality. The statistical results of Unigene length distribution are as follows.’

' DEG的功能注释和注释的差异表达基因(DEGs)数量的统计表。当Unigene同时对三个或更多不同的数据库进行注释时,Unigene注释的准确性得到了证明。

'基于Illumina测序的DEGd数统计表,包括总基因数和上调调控基因数。

“维恩图显示了不同温度组和对照组之间的DEG。

“不同温度组和对照组DEGs的热图,热图中红色表示高表达,蓝色表示低表达。(a) T12 与 T20;(b) T16 与 T20;(C) T24 与 T20;(D) T28 与 T20'

'不同温度组和对照组之间DEGs火山图,红点上调基因,绿点下调基因,蓝点表达无明显变化的基因。(a) T12 与 T20;(b) T16 与 T20;(C) T24 与 T20;(D) T28 与 T20'

' RT-qPC 验证 RNA-Seq 结果以确保转录组数据的准确性。(*P < 0.05, **P < 0.01, ****P < 0.001)'

要点9 

  1. 关于结论,作者应该更加小心,因为他们的数据仅使用肌肉组织产生,因此他们应该仅参考不同治疗对肌肉组织的影响。其他组织对于定义更通用的结论也很重要。结论需要仔细重写。

回应9稿件中的结论已被修改。“我们发现,在温度变化过程中,肌肉的生理代谢受到显着影响。这项工作的发现为未来研究东方暗晦的温度适应提供了基本的肌肉转录组数据。

点 10第 121 – 128 行:您能定义每种治疗方法吗?7行段落仅列出了没有数字就无法理解的数字。表5对此作了更好的描述。

答复1 0这些组在手稿的2.1中定义。

在手稿中进行了修订。“当检测T20和T12组之间不同表达的基因(DEGs)时,结果表明9202个基因差异表达,其中4654和4548个基因被认为是上调和下调的。T4959和T20组间有16个基因差异表达,其中2983个和1976个基因被认为是上调和下调的。T133和T20组间有24个基因差异表达,其中17个和116个基因被认为是上调和下调的。878个基因在T20和T28组之间被证明存在差异表达,其中343个和535个基因被认为是上调和下调的。

11 点:第 132 – 134 行此信息更适合图例。

响应 11此信息已单独放置在图例中。

“不同温度组和对照组之间的DEG火山图,红点上调基因,绿点下调基因,蓝点表达无明显变化的基因。(a) T12 与 T20;(b) T16 与 T20;(C) T24 与 T20;(D) T28 与 T20'

“不同温度组和对照组DEGs热图,热图中红色”表示高表达,蓝色表示低表达。(a) T12 与 T20;(b) T16 与 T20;(C) T24 与 T20;(D) T28 与 T20'

要点 12:第 235 行:删除“_”。

响应 12:“_”已被删除。

以下是我们没有列出附件中标记的更改的一些地方。

我们感谢编辑和审稿人的热情工作,并希望更正得到批准。

再次非常感谢您的意见和建议。

Reviewer 3 Report

The aim of this manuscript is to use high-throughput sequencing technology to analyze 60 days of breeding under five temperature conditions, providing significant evidence to elucidate the mechanisms of temperature acclimation in A. fasciatus.

This manuscript shows rich content, providing a deep insight for some works: the study is within the journal’s scope, and I found it to be well-written, providing sufficient information. Even if the manuscript provides an organic overview, with a densely organized structure and based on well-synthetized evidence, there are some suggestions necessary to make the article complete and fully readable. For these reasons, the manuscript requires major changes.

Please find below an enumerated list of comments on my review of the manuscript:

INTRODUCTION:

LINE 28: The authors should rephrase the sentence as following: “Acrossocheilus fasciatus (Steindachner, 1892) which belongs to Cypriniformes, Cyprinidae, Barbinae, Acrossocheilus, is a lukewarm water fish”.

LINE 42: Furthermore, several transmissible diseases, responsive to antibiotic treatments, have been also investigated in recent years. For this reason, their diagnosis was mainly based on clinical, histopathological, molecular, and genetic evidence (see, for reference: Orioles, M., Galeotti, M., Saccà, E., Bulfoni, M., Corazzin, M., Bianchi, S., ... & Schmidt, J. G. (2022). Effect of temperature on transfer of Midichloria-like organism and development of red mark syndrome in rainbow trout (Oncorhynchus mykiss). Aquaculture560, 738577). This introductive section of the manuscript will benefit from highlighting the pivotal role, played by multidisciplinary techniques, in the study of the growth, development, physiology and pathology of fishes.

LINE 59: Please, remove the term “particular” and use “specific”.

LINE 64: Moreover, RNA-seq is widely used to monitor alterations in the transcriptome of targeted tissues, including new transcripts, splice junction and gene annotations (see, for reference: Wang, L., Wang, B., Hu, C., Wang, C., Gao, C., Jiang, H., & Yan, Y. (2023). Influences of chronic copper exposure on intestinal histology, antioxidative and immune status, and transcriptomic response in freshwater grouper (Acrossocheilus fasciatus). Fish & Shellfish Immunology, 108861). As this study includes RNA-seq evidence, the authors should provide a brief introduction on RNA-seq applications.

The main topic is interesting, and certainly of great clinical impact. As regards the originality and strengths of this manuscript, this is a significant contribute to the ongoing research on this topic, as it extends the research field on the mechanisms of temperature acclimation in A. fasciatus. Overall, the contents are rich, and the authors also give their deep insight for some works.

As regards the section of methods, there is a specific and detailed explanation for the methods used in this study: this is particularly significant, since the manuscript relies on a multitude of methodological and statistical analysis, to derive its conclusions. The methodology applied is overall correct, the results are reliable and adequately discussed.

The conclusion of this manuscript is perfectly in line with the main purpose of the paper: the authors have designed and conducted the study properly. As regards the conclusions, they are well written and present an adequate balance between the description of previous findings and the results presented by the authors.

In conclusion, this manuscript is densely presented and well organized, based on well-synthetized evidence. The authors were lucid in their style of writing, making it easy to read and understand the message, portrayed in the manuscript. Besides, the methodology design was appropriately implemented within the study. However, many of the topics are very concisely covered. This manuscript provided a comprehensive analysis of current knowledge in this field. Moreover, this research has futuristic importance and could be potential for future research. However, major concerns of this manuscript are with the introductive section: for these reasons, I have major comments for this section, for improvement before acceptance for publication. The article is accurate and provides relevant information on the topic and I have some major points to make, that may help to improve the quality of the current manuscript and maximize its scientific impact. I would accept this manuscript if the comments are addressed properly.

Minor editing of English Language are necessary.

Author Response

回复审稿人 2 评论

要点 1: 一 引言

第28行:作者应将句子改写如下:“Acrossocheilus fasciatus(Steindachner,1892)属于Cypriniformes,Cyprinidae,Barbinae,Acrossocheilus,是一种温水鱼”。

响应 1: 它已被修改。Acrossocheilus fasciatus(Steindachner,1892)属于Cypriniformes,Cyprinidae,Barbinae,Acrossocheilus ,是一种温水鱼。

要点 2:介绍

LINE 42:此外,近年来还研究了几种对抗生素治疗有反应的传染性疾病。出于这个原因,他们的诊断主要基于临床,组织病理学,分子和遗传学证据(参见,供参考:金莺,M.,Galeotti,M.,Saccà,E.,Bulfoni,M.,Corazzin,M.,Bianchi,S.,...和施密特,J. G. (2022)。温度对虹鳟鱼中氯样生物转移和红标综合征发展的影响。水产养殖,560,738577)。手稿的这一介绍性部分将受益于强调多学科技术在鱼类生长、发育、生理和病理学研究中发挥的关键作用。

响应 2:文中补充的“同居模型表明,水温的变化可能会影响鱼类养殖的病原体转移和疾病发展,同样面临病原菌和胁迫带来的重大疾病问题。因为水温的变化可能会影响同居模型中疾病和病原体的传播。

特点3引言

第59行:请删除“特定”一词,并使用“特定”。

回应3这在文章中有所更改“转录组是指特定细胞转录的所有RNA的总和”

特点4引言

第 64 行:此外,RNA-seq 广泛用于监测靶组织转录组的改变,包括新的转录本、剪接连接和基因注释(参见,供参考:Wang, L., Wang, B., Hu, C., Wang, C., Gao, C., Jiang, H., & Yan, Y. (2023)。慢性铜暴露对淡水石斑鱼肠道组织学、抗氧化和免疫状态以及转录组反应的影响。鱼类和贝类免疫学,108861)。由于这项研究包括RNA-seq证据,作者应简要介绍RNA-seq的应用。

回应4该文章已被添加,它被广泛用于监测靶组织转录组的改变,包括新的转录本,剪接连接和基因注释'

以下是我们没有列出附件中标记的更改的一些地方

我们感谢编辑和审稿人的热情工作,并希望更正得到批准。

再次非常感谢您的意见和建议。

Round 2

Reviewer 1 Report

I thank the author for the thorough revisions. This paper can be recommended for publication.

Author Response

Response to Reviewer 1 Comments

Thanks for your affirmation.

We appreciate for Editors and Reviewers warm work earnestly.

Once again, thank you very much for your comments and suggestions.

Reviewer 3 Report

The authors have significantly improved the manuscript. I accept for the publication.

Author Response

Response to Reviewer 3 Comments

Thanks for your affirmation.

We appreciate for Editors and Reviewers warm work earnestly.

Once again, thank you very much for your comments and suggestions.